# A Community-Based Therapeutic Education Programme for People with Alcohol Use Disorder in France: A Qualitative Study (ETHER)

**DOI:** 10.3390/ijerph19159228

**Published:** 2022-07-28

**Authors:** Marie Costa, Tangui Barré, Saskia Antwerpes, Marion Coste, Morgane Bureau, Clémence Ramier, Gwenaelle Maradan, Olivier Riccobono-Soulier, Stéphanie Vassas-Goyard, Danielle Casanova, Patrizia Carrieri

**Affiliations:** 1Aix Marseille University, INSERM (Institut National de la Santé et de la Recherche Médicale), IRD (Institut de Recherche pour le Développement), SESSTIM (Sciences Économiques & Sociales de la Santé & Traitement de l’Information Médicale), ISSPAM (Institut des Sciences de la Santé Publique), 27 Bd Jean Moulin, 13385 Marseille, France; tangui.barre@inserm.fr (T.B.); s-antwerpes@web.de (S.A.); marion.coste@inserm.fr (M.C.); morgane-diane.bureau@inserm.fr (M.B.); clemence.ramier@inserm.fr (C.R.); gwenaelle.maradan@inserm.fr (G.M.); maria-patrizia.carrieri@inserm.fr (P.C.); 2Association Addictions France, 84000 Avignon, France; olivier.riccobono-soulier@addictions-france.org (O.R.-S.); danielle.casanova@addictions-france.org (D.C.); 3Association Addictions France, 04000 Digne-les-Bains, France; stephanie.vassas-goyard@addictions-france.org

**Keywords:** alcohol use disorder, harm reduction, therapeutic patient education, controlled drinking, qualitative method, thematic analysis

## Abstract

Therapeutic patient education (TPE) aims to help people with chronic disease strengthen their empowerment and psychosocial skills to better manage their condition. Although TPE has great potential for addiction medicine, studies on its benefits for reducing alcohol-related harms and increasing empowerment are sparse. We conducted a qualitative study of people with alcohol use disorder (AUD) who participated in the community-based TPE programme Choizitaconso to assess their perceptions and experiences of it. Semi-structured interviews were conducted with 16 participants who had completed the TPE programme at least six months previously. The interviews were transcribed and analysed using a sequential thematic analysis. We identified four general themes: (1) the context of participation: the TPE programme could be a strategy to facilitate engagement in AUD care; (2) representations and experiences: the programme helped to “normalize” participants’ relationship with alcohol use by increasing empowerment; (3) TPE strengths: improved knowledge about alcohol use, self-image, weight loss, self-stigma reduction; (4) TPE limitations: difficulty putting learning into practice after the programme ended. The Choizitaconso programme met participants’ health and psychosocial expectations, strengthening their empowerment and reducing self-stigma, thereby facilitating engagement in AUD care.

## 1. Introduction

Unhealthy alcohol use represents a considerable burden for society. It is estimated to be responsible for 20% to 50% of occurrences of cirrhosis, epilepsy, poisoning, road traffic accidents, violence and several types of cancer [1]. Europe has the world’s highest level of alcohol consumption, which is one of the major drivers of liver-related morbidity and mortality [2]. In France, while alcohol use disorder (AUD) prevalence has been evaluated at 10% in the general adult population [3], only 10% of people with AUD (PWAUD) receive related medical care [4].

In the past, standard treatment for patients with alcohol dependence was based on ending consumption, maintaining abstinence, preventing chronic complications of unhealthy consumption, and managing withdrawal symptoms. Besides denial, which is one of the main reasons why PWAUD seek medical care, many PWAUD do not feel ready to quit immediately. In this context, alcohol harm reduction, including reduced daily consumption, has become an acceptable outcome for some PWAUD and care providers [5,6]. This is reflected in many national and international recommendations, where reduced alcohol consumption is considered an alternative goal to abstinence. Several studies have highlighted that AUD care modalities focusing on reduction provide satisfying long-term outcomes in terms of a reduction in alcohol-related harms [7,8].

Patient education, also known as therapeutic patient education (TPE) is a process which aims to help people living with chronic disease gain and maintain hands-on skills to better manage their condition [9]. A patient-centred, multidisciplinary approach includes providing materials and information to patients and their families concerning the condition and how they can manage it. Numerous studies have shown that TPE is effective in preventing complications and in improving both quality of life and treatment adherence for various chronic diseases, including diabetes, asthma and cardiovascular disease [3].

In the context of AUD, TPE programmes can serve as an alternative or complementary strategy to facilitate PWAUD engagement in care and reduce alcohol-related harms and drinking. However, very little research to date has explored the benefits of TPE for this disorder. European examples of such programmes are Alcochoix+, a Canadian programme adopted in Switzerland, and the Körkel [10] model in Germany. Patients enrolled in these programmes reduced their alcohol intake by half and their quality of life improved [11].

Choizitaconso is an ongoing repeated French community-based TPE programme designed to provide people with stronger psychosocial skills so that they can (re)gain control over their alcohol consumption, reduce associated harms, reduce stigma and increase empowerment.

We performed a qualitative study using semi-structured interviews to explore participants’ views of and their experience with the Choizitaconso programme. Our specific objectives were to:-Explore the link between individual factors (history of alcohol use, social and family contexts, etc.) and the decision to participate in the programme.-Observe how these individual factors influenced the implementation of the alcohol harm reduction strategies taught.-Highlight Choizitaconso’s strengths and weaknesses.

## 2. Materials and Methods

### 2.1. Description of the Choizitaconso TPE Programme

In 2016, the “Convergence” Centre for Care, Support and Prevention in Addiction (CSAPA), based in Avignon in the south of France, first implemented Choizitaconso, a TPE programme for PWAUD which can be translated as “choose your consumption”. The programme’s primary objective is to reduce alcohol-related harms and to improve health and quality of life by helping PWAUD learn control techniques and develop adaptation and self-observation skills.

The programme lasts 10 weeks and consists of 10 collective workshops, each lasting 90 min. The workshops’ objectives and themes are presented in Figure 1. Participants also attend several individual meetings with medical staff before, during and after the programme (see Figure 1). Every care provider involved in the programme is trained in TPE and in care for substance use disorder.

### 2.2. Choizitaconso’s Objectives

Choizitaconso aims to:-Promote patient empowerment and deconstruct feelings of self-stigma-Help patients to:
(i)understand the mechanisms which provoke alcohol craving(ii)stop losing control over their alcohol consumption(iii)identify biological causes of AUD, with a view to helping them identify personal risk factors(iv)identify their own perceptions of AUD(v)acquire techniques to control their alcohol consumption(vi)learn to identify internal and external influences on alcohol consumption and at-risk situations(vii)learn to identify the impact of feelings on their behaviour(viii)assess their own expectations and desired effects of alcohol, by developing self-observation skills.

### 2.3. Study Design

ETHER is a longitudinal, multicentre, regional study which started in France in 2019 [12]. Its primary aim is to quantify the impact of the Choizitaconso TPE programme—in terms of alcohol-related harm reduction—on PWAUD at least six months after they finish the programme. Its secondary aim is to design and implement an acceptable and validated evaluation tool for alcohol-related harm reduction adapted to the French AUD context. The study uses a mixed-methods approach which comprises (1) a qualitative study based on semi-structured interviews with former participants of the TPE programme, and (2) a controlled quantitative study which compares the intervention group to a control group not participating in the programme who receive either outpatient or inpatient care.

The study described here aimed to assess ex-participants’ views of and experiences with the Choizitaconso programme, using results from the qualitative component of ETHER.

### 2.4. Objectives of the Qualitative Study

The study described here aimed to assess ex-participants’ views of and experiences with the Choizitaconso programme, using results from the qualitative component of ETHER.

The specific research questions were:-What is the link between individual factors (history of alcohol use, social and family contexts, etc.) and the decision to participate in the Choizitaconso programme?-How do these individual factors influence implementation of the alcohol harm reduction strategies taught in Choizitaconso?-What are Choizitaconso’s strengths and weaknesses?

### 2.5. Method

The study coordinator (DC) presented the qualitative study to potential participants and invited them to participate. Contact information (first name and phone number only) for people who agreed was sent by email to the qualitative study investigator, who then called these potential participants to make an appointment.

Semi-structured interviews took place in a private office at the “Convergence” CSAPA in Avignon to ensure participant confidentiality. MC (Female, 34, doctor of public health, Master of Arts in anthropology) and SA (Female, 24, PhD candidate in Public Health, MA in psychology) were the qualitative study investigators who separately conducted the interviews.

Before each interview, the participant and DC signed the consent and information form. Interviews followed the interview guide presented in Table 1 and were recorded. The guide was created using both existing literature and study objectives. In addition to the themes to be discussed, dunning questions (e.g., Why? What does that mean?) were used to encourage the participants to give more in-depth information in their discourses.

### 2.6. Study Population

The study was conducted in people who had completed the TPE programme at least six months before the interview. Participants could either be receiving treatment or not at the moment of their participation in the study.

### 2.7. Inclusion Criteria

Inclusion criteria were as follows: at least 18 years old, able to provide written, informed consent, fluent French speaker, and completed the TPE programme at least six months before the interview. Exclusion criteria were as follows: being a legally protected adult (under tutorship, curatorship), already participating or planning to participate in another study during ETHER’s six-month follow-up period, and having severe cognitive impairment or a psychiatric disorder.

### 2.8. Analysis

The recorded interviews were transcribed by an external provider.

A manual sequenced thematic analysis [13] was performed on the transcripts as follows:

Step 1:

First, MC and SA read the whole corpus several times. They then individually coded the corpus using Nvivo 1.3.

Step 2:

MC and SA discussed their results in triangulation sessions and agreed on a thematic framework to analyse the corpus.

Step 3:

MC performed the final coding.

## 3. Results

### 3.1. General Characteristics of Study Participants

Table 2 presents the general characteristics of the study participants. Of the 16 participants, nine were women (56%). Participants were born between 1950 and 1985. Over half (62.5%) were born before 1961 and were, therefore, 59 years old or older at the time of the interview. The other characteristics collected concerned participants’ employment and marital status.
a = 1,(1)

### 3.2. Results of Sequential Thematic Analysis

The sequential thematic analysis identified four general themes: (Section 3.2.1) the context of participation: the TPE programme could be a strategy to facilitate engagement in AUD care; (Section 3.2.2) representations and experiences: the programme helped to “normalize” participants’ relationship with alcohol use by increasing empowerment; (Section 3.2.3) TPE strengths: improved knowledge about alcohol use, self-image, weight loss, self-stigma reduction; (Section 3.2.4) TPE limitations: difficulty putting learning into practice after the programme ended. These four themes are described in detail below. These results show the diversity of themes that emerged during the interviews. The verbatim reports are from all the participants in the study.

#### 3.2.1. The Context of Participation

(a)Reasons for participation

Although all participants had AUD (whether prior or current) at programme enrolment, their reasons for participating in Choizitaconso differed. Family was very often mentioned as an element encouraging participants to seek care:


*“It’s the family; it’s the relatives, who are the trigger. Because yes, I think we are aware, well, I was aware that I drank too much, but I didn’t think that it had so much importance and impact on my children, on my husband, on everything. And one day when it got really bad I said “Right. That’s it. I’ll have to”… acting on my own was complicated, so I spoke to my GP first; that seemed to be the most logical solution. He doesn’t deal with AUD so he sent me here.”*
Participant 1


*“… well, I came to the CSAPA after my divorce, so I had no problem with alcohol consumption before; and it wasn’t my divorce in fact, it was the decision about custody over my son that made me drink. I look after my son a lot and I didn’t accept the decision. I got to see him every other weekend whereas [before] I took care of my son all the time; and that didn’t go down well with me, that my son was taken away from me.”*
Participant 12

For others, it was the awareness of a loss of control, an acceleration in alcohol consumption, or difficulty in changing consumption that pushed them to care:


*“… I used to have chronic consumption because of the life I had before and which was difficult; it’s difficult to change; there was a kind of habit, of anxiety about things that were quite present, so that’s why I came here.”*
Participant 6


*“So, I followed the programme; it was on the suggestion, of course, of Doctor C. and her assistants—who are psychologists, nurses, social workers, I think, or educators—like, that I had seen before, and I had come to see them because I was having problems as I kept drinking more and more (…)”*
Participant 8

Finally, health-related concerns were also mentioned as a driving force to seek care, whether it was the patient’s own observation that their health had already deteriorated, or concern about future complications:


*“…I was very worried about my health, but mostly my mental health.”*
Participant 11

Programme expectations and apprehension

The expectations of the majority of people entering the programme were that they would regain control of their drinking, stop drinking completely, or just break habits.

Sometimes the programme content led participants to change their initial expectation (i.e., abstinence or alcohol reduction):


*“Interviewer: When you started this programme, what was your objective? Respondent: to control my consumption. Not abstinence but to control my consumption.”*
Participant 16


*“That’s why I came, to try and find help, to help me break this thing that I myself started, but which over time has become ritualised, and is now very deeply installed in my head.”*
Participant 5


*“My goal is to get back to the age of forty when I hadn’t drunk at all since I was young.”*
Participant 13

Participant 10 had already been treated for alcohol abuse and was already abstinent before starting the programme. The reasons he gave for joining Choizitaconso were to stay vigilant and to learn more about himself:


*“Not expectations, as I was telling you; it was to stay in touch with the disease and with, not to forget and to maintain vigilance over it, and this vigilance is precisely to talk about it and to continue to learn about oneself (…)”*
Participant 10

(b)Alcohol-related representations

Representations of alcohol, AUD and AUD care among people directly concerned by AUD may be the cause of self-stigmatising behaviour or apprehension about treatment. These include the definition of “alcoholism”.


*“I have the impression that people think that being an alcoholic is when you are totally wasted, or that an alcoholic is the local drunk with a bottle of plonk. I think that from the moment you drink alcohol, you are an alcoholic, but there are several degrees, that’s for certain.”*
Participant 13


*“(…) I had a problem with alcohol compared to others who stop, because I think that’s really the difference. When you’re with friends, you can see those who can stop, because the way they function makes it that they’re not alcoholics, and there’s this kind of protection if someone decides “Right. I’m stopping”. And then there’s the one who has a problem with alcohol, who has another drink or two.”*
Participant 3

Regarding the experience of a support group, for many, these are similar to methods such as Alcoholics Anonymous as they are represented in the media. Accordingly, some participants, as we shall see below, were surprised not to be asked to tell their story.


*“(…) we always have these examples of Alcoholics Anonymous meetings, this and that, and everything, where people talk, but that’s in the movies, that’s on TV.”*
Participant 7

(c)Previous or parallel care experiences in Choizitaconso

Most participants had entered the Choizitaconso programme as another element of a complex care pathway which they had already begun. Participants perceived their involvement as a break with negative past experiences of care, which some judged to be useless and even harmful. These previous experiences included stays in inpatient withdrawal units or pharmacological treatments:


*“I can’t remember. Honestly, it was another organisation that was a bit similar to the CSAPA, but I didn’t… I didn’t like it… or it was… so I didn’t continue and I stopped.”*
Participant 10


*“(Selincro) is atrocious. I came back and I told the doctor «No, no way”; So, to avoid the desire to drink, to use it, it works very well. But it’s to consume everything. It’s impossible to eat, I wasn’t hungry any more, I felt like I was drained of all my energy.”*
Participant 1


*“You’re locked up, and then there’s everything, it’s not based on addiction, it’s: you arrive there, you’re sealed up, my mother, the first time she came to see me she said: “get my son out of there”.*
Participant 11

Some participants had previous positive experiences of care, but had subsequently returned to alcohol abuse. This relapse had led them to consider other approaches to tackle their problem. For others, at other times, even if the approach seemed suitable to them, they themselves were not ready to receive care:


*“(…) I consulted a person, but I think it wasn’t really, how can I put it, I think I wasn’t ready. I wasn’t ready; it was a person who seemed qualified, I’m not questioning the professional.”*
Participant 3


*“(…) there are a lot of workshops, (…). You see a psychologist once a week, a psychiatrist, there is no time limit, and you have other people following you all the time, and everyone is up to speed with the case (…). It’s very professional (…) but at the moment, I don’t know, it’s a difficult period.”*
Participant 12

Participants all received parallel support (i.e., outside the Choizitaconso programme) at the same CSAPA in Avignon, in the form of individual consultations or other workshops. They considered that this support complemented the TPE programme.


*“(…) I continue to work with the doctor, who’s more in touch with real life, let’s say, and also more adapted to each person’s case, (…) when it’s really a psychological problem it’s better to treat it individually. So I think these two things were complementary.”*
Participant 5


*“Respondent: (…) yes, so there was relaxation for example… Interviewer: In the Choizitaconso programme, there is a relaxation component? Respondent: ah okay, no, no. That’s another thing. I was talking about everything that was offered here.”*
Participant 10

#### 3.2.2. Representations and Experiences

All of the participants were very enthusiastic about Choizitaconso and described numerous benefits, which we classified under four sub-themes: pedagogical benefits, skills acquisition and empowerment, deconstructing prejudice, and harm reduction.

(a)Pedagogical benefits

The TPE programme has several didactic elements aimed at informing participants about alcohol and related harms. Respondents found these elements to be interesting and useful. They also noted that these elements were absent from other care approaches, leaving participants with a rather passive attitude about the care provided:


*“Because for me, it was, how can I put it, really the first courses; I think it’s complete, in the sense that they really explain things to you. It’s here that I learnt what the quantity of alcohol represents, the different quantities of alcohol, in relation to one drink, two drinks, what that represents, the values [i.e., different doses, etc.] of alcohol, the dangers, the disease, the different dangers; I think it’s much more complete. (…) I know what I have left, how can I put it, my thoughts, when I went to other establishments, and all that, I said to myself that I had learned quite a lot here.”*
Participant 14

(b)Skills acquisition and empowerment

Skills acquisition was the most frequent sub-theme in our coding matrix. By skills acquisition, we mean behavioural changes—for example, increased self-observation—which participants were able to implement after they finished the programme thanks to their participation in it. These skills covered a variety of areas, for example assertiveness and understanding one’s relationship with alcohol, and always resulted in the implementation of meaningful and sustainable behavioural changes in participants’ management of alcohol consumption, leading to a form of empowerment.


*“Something else that is also very important is that we learned to be able to observe ourselves.”*
Participant 11


*“There was a session; it was a video that lasted maybe three minutes; in that video there were 15 s and those 15 s made me understand why I drank alcohol and I understood why I started; in 15 s, after spending three years trying to understand.”*
Participant 12

In addition to acquiring skills to manage consumption, respondents positively commented on the the programme’s aim to help participants become more assertive and understand how to express their desire to drink even in front of relatives opposed to it:


*“That too was a big benefit of the centre; learning to, well learning… knowing how to stand up for myself: that is to say, at some point, I’m not three years old anymore, I’m not here to be lectured.”*
Participant 1

(c)Deconstructing prejudice

This sub-theme covered several aspects. As we mentioned in the first main theme above (i.e., the context of participation), participants sometimes came to the programme with many representations about themselves and alcohol in general. Respondents indicated that the programme enabled them to realize, thanks to contact with other people like themselves, that PWAUD are normal. In this way, the programme helped them regain their self-esteem:


*“(…) is it that [i.e., the programme] which also helps to improve this reality of the image we have of ourselves: ‘I am a fragile human being, like him, like him, like her, like her’”*
Participant 11


*“And it wasn’t that at all. I had a completely wrong idea of what it could be, and I found it comforting first of all to see people with me who, for me, were not the ‘alcoholic type’ [participant makes inverted commas sign], who [i.e., alcoholic types] are always quite negative about, their self-image (…) So for me, in terms of self-esteem, it was beneficial.”*
Participant 5

In addition to changes in perceptions thanks to the observation of their peers, Choizitaconso was also a driver of large changes in terms of decreased self-stigma and guilt. Many respondents described that they were relieved by this shift in their own perceptions:


*“The more you come to the CSAPA, the more you’re involved, and the more you can get away from that [i.e., the shame of coming and of alcohol]. I’m no longer ashamed now. I used to come with my workbag, because there’s a pharmacy opposite, and I said to myself ‘People need to think that I’m coming because I’m a medical professional’. Now I’m never ashamed to come to a CSAPA.”*
Participant 14


*“Because guilt damages your self-esteem, that’s it: “yes, I’m a nothing so I drink”. And this idea… I don’t like the word ‘control’, to have the freedom to take it or not, and even if I wanted to get drunk, I could do it without feeling guilty.”*
Participant 11

Finally, the change in representations can concern those linked to the therapeutic objectives (i.e., alcohol reduction or abstinence). This paradigm shift is one of the driving forces behind the development of Choizitaconso, and many of our respondents commented on it, saying that it removed the pressure of the ‘ideal’ of alcohol abstinence;

*“Well, anyway, it’s never dramatic, because I say to myself that I’ve taken two hundred steps forward and now I’ve taken one backwards, so it’s not serious, I’m going to take two hundred more, so it’s not serious. Whereas before, I would take a step backwards, I would drink and I would say to myself: “Well, I’ve screwed up everything, I’m starting again”*.Participant 12


*“My main objective was total abstinence; it didn’t allow me to reach an objective, it simply allowed me to say to myself that it could be a moment of relaxation—the fact of drinking a beer—it could be a moment of well-being and relaxation.”*
Participant 15


*“Ah yes, I don’t know, it happened after three or four sessions. Yes, they said: ‘you can continue, but you can choose’. [participant makes inverted commas sign] I can keep going.”*
Participant 7

(d)Harm reduction

With regard to alcohol harm reduction, many participants considered that they did not run the risk of any particular harm before they entered the programme:


*“Yes absolutely, but that’s it. As for the rest, I didn’t feel, I didn’t feel sick at all. And in fact, I wasn’t [sick]. D. C. had me tested, everything was fine.”*
Participant 8


*“(…) no, that’s something I didn’t feel at all; I didn’t feel before that I was suffering from any pathologies linked to my alcohol consumption, so obviously I don’t feel I’ve reduced the risks by controlling it better. Do you understand what I mean?”*
Participant 1

However, Participant 1 qualified the above statement by indicating she had in fact reduced harms, as she felt more competent and comfortable in her work following the TPE programme:


*“That’s it. Even though I’ve never had a car accident, I’ve never fallen, I’ve never hurt myself, I’ve never spoken badly to anyone; but the fact remains that in the morning, one, your head isn’t the same, and two, you don’t have the same mental alertness.”*
Participant 1

Some participants reported a change in their behaviour, such as drinking less strong alcohol and reducing the quantities, or becoming aware of the quantities consumed in order to reduce them:


*“Yes. But I don’t drink the same things anymore. Before, I used to drink pastis, whisky, strong spirits and now I drink two glass of white wine.”*
Participant 9


*“For example, one thing that’s interesting, is to say, “here, one glass equals this” [i.e., quantity for standard unit] and it’s true, it’s not the glass that you serve yourself at your friends’ house, where you actually drink three glasses…”*
Participant 6

Furthermore, modifying alcohol consumption enabled patients to (re)find the time and energy to engage in sports. Therefore, it contributed to improving their physical and psychological health:


*“I used to do quite a lot of sport; I used to cycle and swim and when I started drinking again, I gave up completely. And then, little by little, I started again.”*
Participant 15

A reduction in alcohol consumption and better sleep quality were also associated with weight loss, and here again, the health benefits were quickly felt:


*“(…) so at the same time I’ve lost weight. I’ve lost 10 kg, so obviously I’m better. And as I’m quite sporty, I go cycling or skiing in the mountains, so yes, 10 kilos less is nice all the same.”*
Participant 3


*“I went from 94 to 82 kg. I lost 12 kg. And my liver feels much better. It’s happy. So it’s perhaps a bit early to give you more consolidated data, I would say. No, I get up much earlier in the morning, I sleep a lot. I don’t have too many problems falling asleep, and I have proper nights. That’s about it, 8 h of sleep is enough for m; before I needed 10, 11, 12 h. I’m already a little less tired, no more cramps, and physically it’s improved.”*
Participant 2

For other respondents, Choizitaconso had a positive effect on the risk of relapse, as described by participant 4, who felt more serene, because she no longer lived in perpetual fear:


*“Even alcohol, I’m not worried anymore. Before I knew that even after four years it could start again. And now I don’t have that feeling anymore. I don’t want to say I’m cured, it’s over, but I’m at peace, I’m not worried.”*
Participant 4

Although the majority of participants did not report an end to at-risk behaviours, for some the programme provided a wake-up call to change their actions and embrace safer behaviour, particularly with regard to road risk:


*“It helped me not be hospitalised, to manage on my own with… not to go back to hospital, whereas before I would have gone maybe a week later to a clinic, but for a month and a half.”*
Participant 12

#### 3.2.3. TPE Strengths

The programme was described in very positive terms by all participants. We classified its strengths into two sub-themes: (1) workshop climate, and (2) content and organization of sessions (i.e., individual and collective).

(a)Workshop climate

The main strength of Choizitaconso, reported by all of the participants, was its collective nature. As mentioned above, contact with peers enabled some respondents to change their self-perception. Furthermore, the sharing of experiences was seen as enriching—enabling people to progress in their own life project—and as a driver of hope:


*“Then I remember, there were… it wasn’t like role-playing but there were quite a few workshops where we put ourselves in other people’s shoes and I found that really interesting. (…) it involved a lot of games between us and a lot of discussions.”*
Participant 16

*“It gives you hope to spend time with people who have experiences, to be able to talk about them too, it’s important”*.Participant 14

Sharing experiences also enabled participants to take a step back and put their own situation into perspective:

*“(…) I don’t have any problems with my wife and children, everyone is fine, I have no worries. It also helped me to put things into perspective and to say to myself… so, this group helped me do that. Helped put things into perspective and to say to myself: ‘well, I’m not too badly off and I don’t have too much to complain about compared to some people who are marked by the nasty things in life. Because that’s kind of what it’s about”*.Participant 7

The overall atmosphere of the workshops was very much appreciated, and participants enjoyed the social aspect:


*“(…) the whole team is really good, so it’s… there was a very good atmosphere as well, it wasn’t at all… how can I put it… dramatic or whatever; it was… even if we weren’t there to have a laugh, but it wasn’t an oppressive atmosphere.”*
Participant 6

For those most isolated from society, the workshops were an opportunity to find a social life again:


*“(…) one thing: it also allowed me to re-socialise because alcohol makes you feel less social. Well, except for people who go to nightclubs or bars or who still manage to work. But for me, teleworking and being at home all the time, I had reached a point where I didn’t see anyone anymore”.*
Participant 2

(b)Content and organization of sessions (i.e., collective and individual)

One of the major qualities of the programme, from the respondents’ point of view, was its suitability. Irrespective of whether participants were already abstinent or in the process of reducing and/or controlling their consumption, all found the workshops to be relevant. Becoming an actor in one’s own drinking therapeutic objective, and not demonizing the image of alcohol, contributed to adherence to the programme and successful outcomes. One of the respondents even reported that Choizitaconso had helped her to stop smoking:


*“(…) so you have the choice to drink or not to drink, so you don’t see it as a forbidden product anymore.”*
Participant 13


*“And what I liked was that we were all in different situations in terms of consumption. There were abstainers, there were those who drank regularly; there were those who used occasionally, so that’s what I liked in fact.”*
Participant 16


*“And then I used the methods taught for alcohol in order not to smoke again. And it worked. Because I would say to myself: “don’t take the cigarette, wait five minutes, it will pass, do something quickly” and that’s what I did and it worked. Think about something else, keep busy, be active. It just doesn’t happen all by itself. I think that in order for it to last, you have to continue using the methods. Yes. I’m very happy with it, it’s all positive”.*
Participant 4

The more practical aspects of the sessions (time, duration, number of workshops) were also considered satisfactory:


*“(…) the schedules were well adapted, the rhythm was well adapted, the speakers, well I have absolutely nothing to say against this place. On the contrary.”*
Participant 5

Moreover, the constraining nature (in terms of logistics and content) of the programme was mentioned as one of its strong points. The need to commit oneself over several weeks and to organise travel were reasons to be serious about the programme and to implement the lessons learned:


*“The fact that you have to organise yourself for a long-term thing, something that is already planned, yes. All this means that you have to keep your commitments.”*
Participant 3

#### 3.2.4. TPE Limitations

Although patients left the programme very satisfied with the experience, there were some criticisms. These were mainly related to (i) the constraining nature of the programme, and (ii) the difficulty to apply lessons and achieve the programme’s objectives.

(a)Constraining nature of the programme

Some of the restrictive elements mentioned were material and practical in nature, specifically the time and money the participants had to invest to come to the workshops:


*“(…) what was difficult for me was the distance. (…) I had difficulty because of the distance: (…) in [name of home town] in the Bouches-du-Rhône. So that’s a 70-km return trip each time. That’s it. It’s the time. During working hours. The distance, the time it takes, and also financially. The cost of travel… I spend 10 € each time. A round trip. So that’s it, you have to have the budget (…)”*
Participant 1


*“It was a big sacrifice because it meant that I had to make appointments every fortnight in Avignon; well it was difficult, believe me. The reason I agreed was because it was important for me, because it required sacrifices in my work. I was under such pressure… even if it was between noon and two-thirty, [i.e., lunch hours] it was too much, and afterwards, by the time I got back for a meeting, my afternoon was practically ruined.”*
Participant 14

Other restrictive elements were linked to the way in which the workshops were run, with some participants judging them to be over-supervised, with little room for personal expression. The “strictness” of the main facilitator was mentioned several times as unpleasant:


*“But there is one thing where actually at certain times I found it a bit too directed. ‘No, no, we’re not going there, that’s where we want…’ [Respondent cites main facilitator]”*
Participant 7


*“(…) yes, with everyone, when we didn’t go in her direction, this lady didn’t like it, and when she was the moderator… she explained something to us that we didn’t understand, and because we didn’t understand it, it made her angry.”*
Participant 13

(b)Difficulty to apply lessons and achieve programme objectives

Finally, one of the major limitations of the programme, as reported by participants, was that it did not necessarily lead them to complete their objectives. They felt that continued work and vigilance was required, even after the programme’s ten sessions:


*“There are things all the same to, there are things to improve, because as I told you earlier (…) there’s always that thing lingering behind like, because alcohol is everywhere unfortunately; even if you’ve no desire to buy it, it’s in your head.”*
Participant 12


*“It’s not perfect, it’s not that, but I think it’s [i.e., alcohol consumption] not a good option. But, like, I haven’t found another way [i.e., something besides alcohol to help] yet. And I’m not strong enough on myself to say no.”*
Participant 1


*“(…) well, it’s always this problem of ritual, until you break it; it’s been there for too long and I find it a bit difficult to change.”*
Participant 5

However, this notion was qualified by participant 6, who indicated that while she did not reach the ultimate goal, the programme helped her take a big step towards reaching it:


*“Yes well, it’s never achieved, but I stuck to what I told myself: no more daily consumption. So, I sometimes drink quite a lot at certain festive occasions. I don’t know if I’ve reached the objective, but I’ve reached one in any case.”*
Participant 6

Finally, the collective nature of the approach implied that not everyone will be able to put the learning into practice, as reported by this participant:


*“And then there were little tools that I had, which I had been given in therapy sessions and that I’d never managed to put into practice, to implement.”*
Participant 2

## 4. Discussion

The main results from the present study, using data from the larger ETHER study, are that the ex-participants reported that the Choizitaconso TPE programme helped them strengthen empowerment, improve knowledge about alcohol use, acquire skills (understood here as behavioural changes), increase self-esteem, reduce self-stigma, and improve family relations. Regaining self-esteem, which has often been damaged by periods of drug use disorders, makes it possible to act in a sustainable way and, for example, to manage an episode of relapse [14]. Our results are in line with those observed in other studies on harm reduction in the field of AUD [15].

The study results tend to indicate that, contrary to what one might think, the openness of the TPE programme to controlled consumption does not constitute an obstacle to participating in it for persons who prefer the objective of total abstinence.

The programme reduces alcohol-related harms, but not in the most common senses of the term (i.e., road risk, work accident, injury, fight, assault, etc.). This is due to the profiles of the people taking part in Choizitaconso: the participants had never been or were no longer at a dangerous stage of their consumption when they entered the programme. Specifically, the harm reduction described by the study’s participants regarded regaining good professional and sporting performance and improved sleep quality, which are important elements for physical and psychological well-being [16,17], contributing to reduce longer-term health risks, particularly for cancer and cardiovascular disease.

Among the programme’s limitations, as perceived by the study’s participants, the most relevant was the difficulty for participants to reach their “ideal” objective. We can, therefore, assume that the programme alone cannot be considered in itself a comprehensive approach for AUD care and will work only when combined with other approaches. This combined prevention strategy can enable people to acquire skills, techniques and know-how in a sustainable way, to move towards better practices, to improve their health, particularly their mental health, and to become responsible actors in their drinking or abstinence rather than adopting a passive attitude, which our participants referred to when talking about previous experiences of coercive therapeutic approaches. Choizitaconso is an illustration of a community-built pedagogic intervention; by offering participants the opportunity to develop and share their experiential knowledge [18] as alcohol consumers, it enables them to strengthen several dimensions of empowerment [19], including the power to decide about one’s alcohol consumption, the power to express it to others, and to choose which therapeutic methods to use from the various support services offered.

Choizitaconso meets participants’ health and psychosocial expectations, specifically increased empowerment and reduced self-stigma, thereby facilitating engagement in AUD care [2]. As community-based interventions targeting alcohol harm reduction can attract more PWAUD to care, this approach should be evaluated as an additional tool in the comprehensive management of this population. In order to better evaluate this intervention in comparison to other AUD treatments, the Choizitaconso programme could be a part of a randomized trial aiming to understand how patient–treatment interactions relate to outcomes. One example of such a trial is MATCH, which assessed how AUD patients’ characteristics (socioeconomic, medical, etc.) helped them to respond more favourably to one treatment than another [20].

### Strengths and Limitations

The primary strengths of the present study are that it is one of the rare studies describing the experience of a community-based (i.e., built with peers) TPE programme for the reduction of alcohol-related harms, and is the first study of its kind in the French context.

In addition, our study allowed us to answer our research questions (i.e., What is the link between individual factors and the decision to participate in the Choizitaconso programme? (Theme 1), How do these individual factors influence the implementation of the alcohol harm reduction strategies taught in Choizitaconso? (Theme 2), What are Choizitaconso’s strengths and weaknesses? (Themes 3 and 4)).

However, study participant confusion between the TPE programme and the other AUD support services provided in the same CSAPA where the Choizitaconso programme is implemented probably constitutes a bias in assessing the programme. In addition, the length of time between programme participation and the date of the study interview may have led to memory bias.

## 5. Conclusions

In the field of AUD care, a personalized treatment model cannot be implemented without paying particular attention to patients’ objectives, and more specifically, to their consumption objectives. It is important, as the French Society for AUD Care emphasized in its latest recommendations, to respect patients’ desires. The viability of controlled drinking as a therapeutic goal remains a subject of debate among AUD care professionals, but previous studies have shown that refusal/fear of abstinence is a major barrier for people considering care to address their alcohol consumption [21]. It has been suggested that the therapeutic goal, whether it is controlled drinking or abstinence, should be an integral part of each patient’s care [22,23]. Moreover, given the French cultural context, where alcohol consumption—in particular, daily wine consumption—is a cultural reality, it is essential to consider options other than total abstinence to encourage people with AUD to enter care.

Harm reduction addresses these cultural issues by offering easily accessible, non-stigmatising and flexible treatment options with a variety of treatment methods, goals and approaches tailored to each individual’s needs. Treatment, whatever its objective (reduction, abstinence, etc.), is better than no treatment [24]. In terms of AUD, harm reduction offers a pragmatic approach to prevent AUD, as it shifts the focus from alcohol consumption to the underlying mechanisms and consequences.

Harm reduction interventions could be offered to other populations that are not ready to engage in treatment—for example, students who binge drink [25]. It can provide incentives (e.g., discussing the negative consequences the person is experiencing) that motivate their desire to change their level of consumption.

Prevention efforts should continue to focus on developing skills and reducing the various risks associated with alcohol use, while treatment approaches should integrate goal setting (i.e., reduction or abstinence) and lifestyle considerations within the broader context of an individual’s social, physical and psychological environment.

## Figures and Tables

**Figure 1 ijerph-19-09228-f001:**
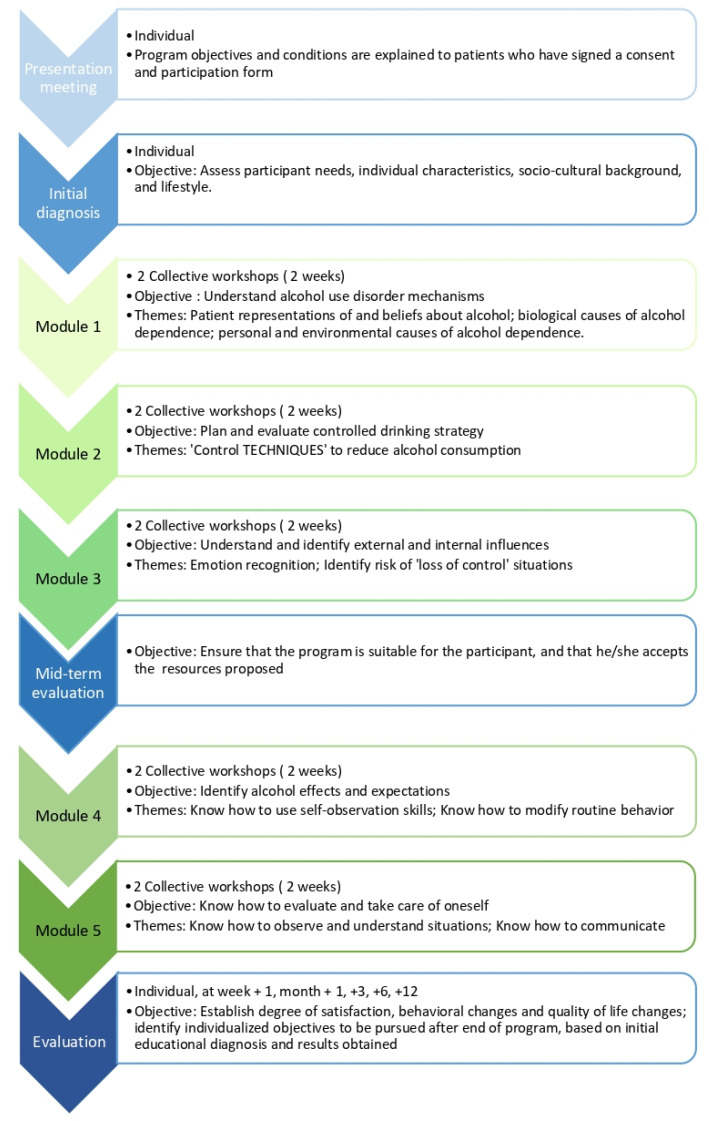
Programme presentation.

**Table 1 ijerph-19-09228-t001:** Semi-structured PWAUD interview guide.

Semi-Structured PWAUD Interview Guide	
**Opening question**	Can you tell me about your experience with the programme Choizitaconso?
**If the interviewed person does not mention these topics spontaneously, the interviewer must do so.**	➢ Why did you decide to participate in this programme? ➢ What were your goals and expectations? ➢ In your opinion, what are the strengths and weaknesses of this programme? ➢ How did you feel during the workshops? Did you ever feel uncomfortable? ➢ Now that the programme is over, do you manage to put into practice what you learned? ➢ How do you see the future of your alcohol consumption?

**Table 2 ijerph-19-09228-t002:** General characteristics of participants.

Participant	Age (Years)	Gender	Employment Status	Marital Status
**01**	45–49	Female	Employed	Married with child(ren)
**02**	65–69	Male	Employed	Living alone with no child
**03**	50–54	Male	Retired	Married
**04**	65–69	Female	Retired	Living alone with child(ren)
**05**	50–54	Female	Employed	Married with child(ren)
**06**	74–79	Male	Retired	Living alone with no child
**07**	55–59	Male	Retired	Married with child(ren)
**08**	74–79	Female	Retired	Married with child(ren)
**09**	64–69	Female	Housewife	Married with child(ren)
**10**	65–69	Male	Retired	Married with no child
**11**	50–54	Female	Employed	In a relationship, with no child
**12**	31–35	Male	Employed	Divorced with child(ren)
**13**	55–59	Female	Employed	Living alone with no child
**14**	31–35	Female	Unemployed	In a relationship, with child(ren)
**15**	55–59	Male	Unemployed	Widowed with no child
**16**	55–59	Female	Training	Separated with child(ren)

## Data Availability

The datasets generated during and/or analyzed during the current study are available from the corresponding author on reasonable request.

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
