# Peer review of "A Community-Based Therapeutic Education Programme for People with Alcohol Use Disorder in France: A Qualitative Study (ETHER)"

_ijerph, 2022, doi:10.3390/ijerph19159228_

Round 1

Reviewer 1 Report

Dear Authors,

this study is interesting and important for practics, but needs to be improved:

1.         In the part of participants: the authors should add a inclusion and exclusion criteria of patients.

2.         In the part of participants: the authors should describe: Where were patients treat? In the out-patient or in-patient clinic?

3.         In the part of participants the authors should describe of diagnosis criteria AUD, which was used during treatment. It is important because of alcohol dependence symptoms include "loss contol of alcohol drinking". Some patients might have "sens of control" and others not.

4.         In the part of „Methods”: authors should write more about the ETHER programme (how ETHER is conected with programme used in this study).

5.         In the part of „Abstract” and „Methods”:  in the abstract, authors described:  „The interviews were transcribed and analysed using a sequential thematic analysis. We identified four general themes: (1) the context of participation: the TPE programme could be a strategy to facilitate engagement in AUD care; (2) Representations and experiences: the programme helped to “normalize” participants’ relationship with alcohol use by increasing empowerment; (3) TPE strengths: improved knowledge about alcohol use, self-image, weight loss, self-stigma reduction; (4) TPE limitations: difficulty putting learning into practice after the programme ended”

According to this text, I see that the authors conducted two the interview (before and after the therapy). It should be described both interviews in the part of „Methods”.

6.         In the part of "Results": the authors should add note that described answers are from selected patients and Why is it?

7.         In the part of „Results” (There, where it is possible) the authors should describe percentage positive and negavite answers. For a better show of results.

8.         Lines 546-549 Is this text it limitations of study? If „yes”, authors should write headline „Limitations of study” in this place.

9.         Could authors describe a relationship between project MATCH and conclusions this studies?  (Project MATCH (Matching Alcoholism Treatment to Client Heterogeneity): rationale and methods for a multisite clinical trial matching patients to alcoholism treatment DOI: 10.1111/j.1530-0277.1993.tb05219.x).

Best regards

Reviewer

Author Response

Dear Editor and reviewers,

Thank you for reviewing our manuscript ijerph-1821638. Your comments have been very useful to help us improve our article. Please find all the changes we have made below. We hope that this new version of the manuscript will meet the requirements for publishing in the International Journal of Environmental Research and Public Health. We remain at your disposal for any further clarifications you may need.

Reviewer 1

Dear Authors,

This study is interesting and important for practics, but needs to be improved:

  1. In the part of participants: the authors should add a inclusion and exclusion criteria of patients.

We have added this information as follows (p6):

“Inclusion criteria

Inclusion criteria were as follows: at least 18 years old, able to provide written, informed consent, fluent French speaker, and completed the TPE programme at least six months before the interview. Exclusion criteria were as follows: being a legally-protected adult (under tutorship, curatorship), already participating or planning to participate in another study during ETHER’s six-month follow-up period, and having severe cognitive impairment or a psychiatric disorder.”

  1. In the part of participants: the authors should describe: Where were patients treat? In the outpatient or in-patient clinic?

We have added this information as follows (p5):

“Study population

The study was conducted in people who had completed the TPE programme at least six months before the interview. Participants could either be receiving treatment or not at the moment of their participation in the study.”

  1. In the part of participants the authors should describe of diagnosis criteria AUD, which was used during treatment. It is important because of alcohol dependence symptoms include "loss contol of alcohol drinking". Some patients might have "sens of control" and others not.

Thank you for this comment. We did not collect data on the diagnosis criteria for AUD during Choizitaconso, as the present qualitative study’s focus was on assessing ex-participants’ views and experience of the programme. We aim to collect and include these data in the future planned quantitative study.

  1. In the part of “Methods”: authors should write more about the ETHER programme (how ETHER is connected with programme used in this study).

We have tried to clarify this point by creating a new subsection as follows(P5):

“2.4 Objective of the qualitative study

The study described here aimed to assess ex-participants’ views and experience of the Choizitaconso programme, using results from the qualitative component of ETHER.”

  1. In the part of “Abstract” and “Methods”: in the abstract, authors described: “The interviews were transcribed and analysed using a sequential thematic analysis. We identified four general themes: (1) the context of participation: the TPE programme could be a strategy to facilitate engagement in AUD care; (2) Representations and experiences: the programme helped to “normalize” participants’ relationship with alcohol use by increasing empowerment; (3) TPE strengths: improved knowledge about alcohol use, self-image, weight loss, self-stigma reduction; (4) TPE limitations: difficulty putting learning into practice after the programme ended”

According to this text, I see that the authors conducted two the interview (before and after the therapy). It should be described both interviews in the part of “Methods”.

There is a misunderstanding concerning this point. Only one interview tool place; after participants completed the Choizitaconso programme. We clarified this as follows (P5)

“Participants were interviewed once, after their participation in the Choizitaconso programme.”

  1. In the part of "Results": the authors should add note that described answers are from selected patients and Why is it?

We have added a statement in the results section (p7):

“These results show the diversity of themes that emerged during the interviews. The verbatim reports are from all the participants in the study.”

  1. In the part of „Results” (There, where it is possible) the authors should describe percentage positive and negavite answers. For a better show of results.

Thank you for this comment. However, as this study is qualitative (based on semi-structured interviews), we cannot calculate the percentage of positive and negative responses. The interviews aim to document the participants’ experiences and we cannot interpret them as being positive or negative.

  1. Lines 546-549 Is this text it limitations of study? If „yes”, authors should write headline „Limitations of study” in this place.

We have added a subsection “Strengths and limitations” on page 15.

  1. Could authors describe a relationship between project MATCH and conclusions this studies? (Project MATCH (Matching Alcoholism Treatment to Client Heterogeneity): rationale and methods for a multisite clinical trial matching patients to alcoholism treatment DOI: 10.1111/j.1530-0277.1993.tb05219.x).

Thank you for sharing this interesting article with us; we now cite it and discuss it as follows:

“In order to better evaluate this intervention in comparison to other AUD treatments, the Choizitaconso programme could be a part of a randomized trial aiming to understand how patient-treatment interactions relate to outcomes. One example of such a trial is  MATCH, which assessed how AUD patients’ characteristics (socioeconomic, medical, etc.) helped them to respond more favourably to one treatment than another (Project MATCH (Matching Alcoholism Treatment to Client Heterogeneity), 1993)

Best regards

Thank you

Reviewer 2 Report

This is an interesting paper with valuable insight into a much needed area of research.

The abstract introduces the content of the paper appropriately. The introduced theories and previous research results are in line with the content and direction of the paper. However, besides the aims, basic research questions or exact hypotheses should be introduced too, possible at the end of the Introduction section (after introducing the aims). 

Concerning the Methods section, the program, procedure and instruments are described correctly in an understandable and coherent way. However, the characteristics of the sample (General characteristics of study participants) should also be introduced here.

Results are also clear. The introduction of the results follows the group of questions of the interviews. Concerning the format, sometimes it is hard to see which subchapter belongs where, and which quotation belongs to whom.

Discussion and conclusions are correctly defined. However, it should be necessary to reflect on the assumptions of the authors i.e. the research questions of hypotheses. This is now missing, however, research findings are compared to previous research results in a correct manner.

In my opinion, this paper is an intriguing paper introducing extremely important practical information. It is worth publishing this paper.

Author Response

Dear Editor and reviewers,

Thank you for reviewing our manuscript ijerph-1821638. Your comments have been very useful to help us improve our article. Please find all the changes we have made below. We hope that this new version of the manuscript will meet the requirements for publishing in the International Journal of Environmental Research and Public Health. We remain at your disposal for any further clarifications you may need.

This is an interesting paper with valuable insight into a much needed area of research.

The abstract introduces the content of the paper appropriately. The introduced theories and previous research results are in line with the content and direction of the paper. However, besides the aims, basic research questions or exact hypotheses should be introduced too, possible at the end of the Introduction section (after introducing the aims). 

We have added a subsection concerning this point (p5)

“2.4 Objectives of the qualitative study

The study described here aimed to assess ex-participants’ views and experience of the Choizitaconso programme, using results from the qualitative component of ETHER.

The sppecific research questions were:

-           What is the link between individual factors (history of alcohol use, social and family contexts, etc.) and the decision to participate in the Choizitaconso programme?

-           How do these individual factors influence the implementation of the alcohol harm reduction strategies taught in Choizitaconso?

-           What are Choizitaconso’s strengths and weaknesses?”

Concerning the Methods section, the program, procedure and instruments are described correctly in an understandable and coherent way. However, the characteristics of the sample (General characteristics of study participants) should also be introduced here.

Thank you for raising this point; we have added two subsections: “Inclusion Criteria” and “Study Population” to document this point.

Results are also clear. The introduction of the results follows the group of questions of the interviews. Concerning the format, sometimes it is hard to see which subchapter belongs where, and which quotation belongs to whom.

We agree that our presentation was not very clear. We have added line breaks and numbering throughout the document to facilitate reading.

Discussion and conclusions are correctly defined. However, it should be necessary to reflect on the assumptions of the authors i.e. the research questions of hypotheses. This is now missing, however, research findings are compared to previous research results in a correct manner.

Thank you for this point. Now that we have made our research questions more explicit (see response to your first point above), we believe that the link between these questions and our findings is more obvious. We have added a paragraph to explain the relationship between our research questions and our results in the Strengths and Limitations” section (P15) as follows:

“In addition, our study allowed us to answer our research questions (i.e., What is the link between individual factors and the decision to participate in the Choizitaconso programme? (Theme 1) How these individual factors influence the implementation of the alcohol harm reduction strategies taught in Choizitaconso? (Theme 2) What are Choizitaconso’s strengths and weaknesses?” (Theme 3 and 4).”

In my opinion, this paper is an intriguing paper introducing extremely important practical information. It is worth publishing this paper.

Thank you again for your valuable comments and for endorsing the publication of our article.
